



# Technical note: Accelerator mass spectrometry of $^{10}$Be and $^{26}$Al at low nuclide concentrations

Klaus M. Wilcken[1], Alexandru T. Codilean[2,3], Réka-H. Fülöp[1,2], Steven Kotevski[1], Anna H. Rood[4], Dylan H. Rood[4], Alexander J. Seal[4], and Krista Simon[1]

[1]Australian Nuclear Science and Technology Organisation (ANSTO), Lucas Heights, NSW 2234, Australia
[2]School of Earth, Atmospheric and Life Sciences, University of Wollongong, Wollongong, NSW 2522, Australia
[3]ARC Centre of Excellence for Australian Biodiversity and Heritage (CABAH), University of Wollongong, Wollongong, NSW 2522, Australia
[4]Department of Earth Science and Engineering, Royal School of Mines, Imperial College London, London SW7 2AZ, UK

**Correspondence:** Klaus Wilcken (klausw@ansto.gov.au)

**Abstract.**

Accelerator Mass Spectrometry (AMS) is currently the standard technique to measure cosmogenic $^{10}$Be and $^{26}$Al concentrations, but the challenge with measuring low nuclide concentrations is to combine high AMS measurement efficiency with low backgrounds. The current standard measurement setup at ANSTO uses the 3+ charge state with Ar stripper gas at 6 MV for Be and 4 MV for Al, achieving ion transmission through the accelerator for $^{10}$Be$^{3+}$ and $^{26}$Al$^{3+}$ of around 35% and 40%, respectively. Traditionally, $^{26}$Al measurement uncertainties are larger than those for $^{10}$Be. Here, however, we show that $^{26}$Al can be measured to similar precision as $^{10}$Be even for samples with $^{26}$Al/$^{27}$Al ratios in the range of $10^{-15}$, provided that measurement times are sufficiently long. For example, we can achieve uncertainties of 5% for $^{26}$Al/$^{27}$Al ratios around $1 \times 10^{-14}$, typical for samples of late-Holocene age or samples with long burial histories. We also provide empirical functions between the isotope ratio and achievable measurement precision, which allow predictive capabilities for future projects and serve as a benchmark for inter-laboratory comparisons. For the smallest signals, not only is understanding the source of $^{10}$Be or $^{26}$Al background events required to select the most appropriate blank correction method but also the impact of the data reduction algorithms on the obtained nuclide concentration becomes pronounced. Here we discuss approaches to background correction and recommend quality assurance practices that guide the most appropriate background correction method. Our sensitivity analysis demonstrates a 30% difference between different background correction methods for samples with $^{26}$Al/$^{27}$Al ratios below $10^{-14}$. Finally, we show that when the measured signal is small and the number of rare isotope counts is also low, differing $^{26}$Al or $^{10}$Be concentrations may be obtained from the same data if alternate data reduction algorithms are used. Differences in the resulting isotope concentration can be 50% or more if only very few ($\lesssim 10$) counts were recorded or about 30% if single measurement is shorter than 10 min. Our study presents a comprehensive method for analysis of cosmogenic $^{10}$Be and $^{26}$Al samples down to isotope concentrations of few thousand atoms per gram of sample, which opens the door to new and more varied applications of cosmogenic nuclide analysis.



## 1   Introduction

Cosmogenic nuclides are now routinely applied in the Earth sciences (e.g., Granger et al., 2013; Blanckenburg and Willenbring, 2014). Over the last three decades, the technique has revolutionised the field of quantitative geomorphology (e.g., Granger and Schaller, 2014; Dixon and Riebe, 2014) and has made important contributions to the reconstruction of glacier chronologies and past climate changes (e.g., Ivy-Ochs and Briner, 2014; Balco, 2019). While cosmogenic $^{10}$Be has been the workhorse for Earth science applications, $^{26}$Al – as part of the $^{26}$Al/$^{10}$Be nuclide pair – has also been widely applied in studies of burial dating (e.g., Granger, 2006; Balco and Rovey, 2008) and in settings where material has experienced a complex exposure history. The latter includes landforms repeatedly covered by ice (Knudsen and Egholm, 2018, and references therein) and the sediment of large river basins (Fülöp et al., 2020; Wittmann et al., 2020).

Accelerator Mass Spectrometry (AMS) is currently the standard technique to measure cosmogenic $^{10}$Be and $^{26}$Al concentrations and while the majority of routine samples yield $^{10}$Be/$^{9}$Be or $^{26}$Al/$^{27}$Al ratios $\sim 10^{-13}$ or above, more recent applications often push the limits of the technique. Examples of such applications, routinely yielding isotope ratios in order of $10^{-14}$ to $10^{-15}$, include dating of young glacial deposits (Schaefer et al., 2009), using onshore and offshore bedrock and sediment cores to reconstruct the glaciation histories of polar regions (Bierman et al., 2016; Schaefer et al., 2016; Shakun et al., 2018) or to reconstruct paleo-erosion rate records (Lenard et al., 2020; Mariotti et al., 2021; Mandal et al., 2021, see also Codilean and Sadler, 2021), estimating coastal cliff retreat rates (Hurst et al., 2016; Swirad et al., 2020) or denudation rates in rapidly uplifting terrain (Derrieux et al., 2014; Siame et al., 2011), dating of old ice (Auer et al., 2009), and search for supernova signatures (Feige et al., 2018). Furthermore, recent refinements to burial dating models (Knudsen et al., 2020) will likely translate into an increase in isochron burial dating applications in the near future. The latter require accurate and precise determination of $^{26}$Al/$^{27}$Al ratios that are often close to background values.

The above applications are not only pushing the limits of the technique but also require increased sample numbers. The answer to both challenges at some level is to optimise and improve the AMS measurement efficiency. In this study, we quantify the measurement losses that limit the achievable measurement sensitivity for $^{10}$Be and $^{26}$Al analyses at ANSTO's AMS facilities, discuss approaches to background correction for low ratio $^{10}$Be and $^{26}$Al measurements, and discuss approaches to calculating nuclide concentrations from small numbers of rare isotope counts. Although the data presented here is specific to the setup at ANSTO, the conclusions drawn from these data are more widely applicable.

## 2   Accelerator mass spectrometry of $^{10}$Be and $^{26}$Al at ANSTO

The principle behind an AMS measurement, including that of $^{10}$Be and $^{26}$Al, is as follows (Finkel and Suter, 1993; Fifield, 1999): Negative ions are extracted from the sample via Cs sputtering (Middleton, 1983; Southon and Santos, 2007) and after energy and momentum analysis they are injected into an accelerator. In the accelerator terminal the ions undergo collisions with gas molecules (or a solid foil may also be used) that result in molecule break-up and the injected ions becoming positively charged. A second energy and momentum analysis after the accelerator is used to select one mass with a given charge state for the final ion detection.





Recent important technical improvements in the AMS method are summarised by Synal (2013). In the context of $^{10}$Be and $^{26}$Al, these include:

   (i)  molecular breakup at 1+ charge state (Lee et al., 1984; Suter et al., 1997; Synal et al., 2000);

  (ii)  development of He gas strippers (Lachner et al., 2014; Müller et al., 2015);

 (iii)  development of ion detection methods to suppress $^{10}$B interference at low energies (Grajcar et al., 2004, 2007; Müller

60         et al., 2008, 2010);

 (iv)  development of the gas-filled magnet method to enable magnesium suppression for $^{26}$Al measurements using AlO$^-$ ions
        (Paul et al., 1989; Arazi et al., 2004; Fifield et al., 2007; Miltenberger et al., 2017).

    As a result of the above technical improvements, it is now possible to measure $^{10}$Be and $^{26}$Al with accelerators that range from a few hundred kV to 10 MV or larger. In general all these varying measurement methods can achieve backgrounds

that are sufficient for the majority of routine samples with $^{10}$Be/$^9$Be or $^{26}$Al/$^{27}$Al ratios $\sim 10^{-13}$ or above, and have led to increased accessibility with a growing number of AMS facilities that are capable of measuring cosmogenic nuclides. However, a challenge is posed by samples with low nuclide concentrations because the losses during the measurement will directly affect the sensitivity and achievable statistical precision.

    AMS measurement efficiency is characterised by losses in: (i) ion source, (ii) ion transport, (iii) charge state yields, and (iv)

ion detection. Understanding, quantifying, and subsequently minimising these losses will improve the measurement sensitivity and increase the achievable statistical precision for $^{10}$Be and $^{26}$Al analysis. Typically the largest losses arise from the inefficient negative ionisation in the ion source, with lesser but still significant losses in the transmission of the ions through the accelerator and ion detection.

## 2.1  Measurement of $^{10}$Be

Unfortunately beryllium does not form negative ions as readily as carbon, for example, and the achieved ionisation efficiencies for beryllium are much lower. Standard practice is to use molecular BeO$^-$ ions for $^{10}$Be measurement as they are more prolific than elemental negative beryllium ions (Middleton, 1989). Nevertheless, the highest reported ionisation efficiencies using BeO$^-$ are $\sim$3% (Middleton, 1989; Rood et al., 2010; Wilcken et al., 2019), whereas for C$^-$ values between 10% to 30% have been reported (Fallon et al., 2007; Hlavenka et al., 2017; Yokoyama et al., 2010). The poor negative ion yield for Be and Al

(see section 2.2) is sometimes taken to be an inherent limitation of the technique, rather than a challenge to be addressed.

    There is significant scope for improvement in the performance and operation of negative ion Cs-sputter sources, in terms of their stability, consistency, sample consumption and efficiency. However, this is challenging as some design considerations are somewhat contradictory, e.g., to ensure expedient sample consumption the sputtering rate of the material needs to be high which in turn makes it difficult to keep the source insulators clean. The lack of a well-understood theoretical model for

the negative ionisation process further adds to the engineering challenges. Modifications to the source design that improve longevity, stability, optimise the sample consumption, and improve negative ionisation probability, are equally important.





Practical methods to optimise sample consumption and negative ionisation include: (i) optimising the shape of the sample holder, (ii) where the sample is loaded relative to the sample holder, (iii) what binding material is used and how much, (iv) where the sample holder is located within the ion source, and (v) how the ion source is operated. Inconveniently, different ion

source designs require their own optimisations, and different isotopes are likely to behave differently with the same ion source. For example, the optimal recess depth where sample is loaded within the sample holder is different between $C^-$, $BeO^-$ and $Al^-$ ions (Yokoyama et al., 2010; Auer et al., 2007; Hunt et al., 2006, 2007).

The nature of tandem acceleration is that only a certain fraction of the negative ions that are injected to the accelerator go into the positive charge state that is selected for analysis. This is the second most significant loss in the AMS measurement.

Fig. 1 shows the transmission of the injected $BeO^-$ ions through the accelerator and into various charge states as a function of $^9Be$ ion energy at the terminal for Ar and He stripper gases. Selecting the most prominent charge state is an advantage but impacts the ion detection as will be discussed below.

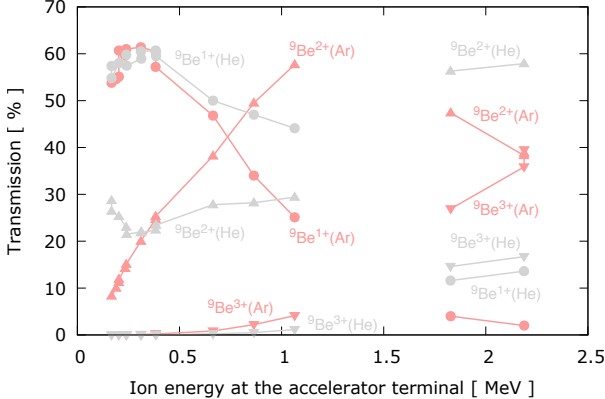

**Figure 1.** Compilation of $^9Be$ transmission measurements on ANSTO's VEGA and SIRIUS accelerators to various charge states as a function of ion energy at the accelerator terminal with helium (grey) and argon (red) gas stripping. The gap in the transmission data represents an energy region where ion optical losses through the accelerator have greater impact on the measured charge states and so these data have been excluded.

Transmission data in Fig. 1 collates measurements on ANSTO's 1 MV VEGA (Wilcken et al., 2015) and 6 MV SIRIUS (Wilcken et al., 2017) accelerators using $Be^-$, $BeO^-$ and $BeO_2^-$ ions to be able to cover a wide energy range. $^9Be$ ion energy

at the accelerator terminal is a function of the accelerator voltage and mass of the molecule injected. Fig. 1 shows that charge state losses can be minimised if 1+ or 2+ ions are selected for analysis with maximum transmission being 60% with Ar and He stripper gases. For 1+, the maximum yield is achieved at ~0.3 MeV (Ar and He), whereas 2+ transmission peaks occur after 1 MeV with Ar but later at ~2.2 MeV with He. We cannot reach the charge state peak for 3+ neither with Ar nor He with the maximum 6 MV acceleration voltage, where transmissions are ~35 % and ~18%, respectively.





Currently, our standard measurement setup uses the 3+ charge state with Ar stripper gas at 6 MV accelerator voltage. This is combined with the conventional passive absorber cell method where the interfering $^{10}$B ions are stopped within the absorber cell in front of the ion detector whilst $^{10}$Be ions pass through (Klein et al., 1982; Raisbeck et al., 1984). The method offers an effective suppression of $^{10}$B (see section 3.1) without compromising the measurement efficiency. The measured raw ratio of a standard material is typically between 80% to 90% of the reference value, which is the result of transmission differences between $^{9}$Be and $^{10}$Be and losses during ion detection. Whilst measurements of $^{10}$Be/$^{9}$Be ratios at the level $10^{-16}$ or below are possible (Wilcken et al., 2019), typically the blank samples are in the range from $3 \times 10^{-16}$ to $3 \times 10^{-15}$. These blank levels do not include any additional boron correction and depend on the carrier used and the sample preparation laboratory where the samples were prepared.

The above measurement setup is preferred over low energy 1+ measurements (Wilcken et al., 2015) because the total measurement efficiency is higher. That is, even though the transmission to 1+ charge state at 1 MV through the accelerator is 60%, which is nearly double what is achieved to 3+ at 6 MV, the losses in the ion detection to suppress $^{10}$B interference in our case mean that the total efficiency is currently a factor of ∼3 lower when using the 1+ instead of the 3+ method.

Recently it has been demonstrated that the high transmission of 55% to 2+ at 3 MV can be used for $^{10}$Be measurements and backgrounds below $7 \times 10^{-16}$ without significant losses in the ion detection efficiency (Steier et al., 2019). However, we have not assessed how the suppression of $^{10}$B or other interferences compares with our current standard method, and hence will not discuss this approach further.

## 2.2 Measurement of $^{26}$Al

Similar to, but worse than, $^{10}$Be, low ionisation efficiency is the major challenge for $^{26}$Al measurements. For $^{26}$Al, one is faced with two options as the measurements can be done by either using Al$^{-}$ or AlO$^{-}$ beams. A distinct advantage of using Al$^{-}$ is that $^{26}$Mg does not form stable negative ions and the measurement of $^{26}$Al becomes free from isobaric interferences. Unfortunately, the highest reported ionisation efficiencies for Al$^{-}$ are around 0.2% (Middleton, 1989; Auer et al., 2007; Wilcken et al., 2017). In contrast, the molecular AlO$^{-}$ beam is typically in the order of 10 – 20 times higher than Al$^{-}$ but ion detection of $^{26}$Al requires suppression of the interfering $^{26}$Mg ions that are injected to the accelerator as $^{26}$MgO$^{-}$. Whilst the separation of $^{26}$Al and $^{26}$Mg can be done using the gas-filled magnet technique (Paul et al., 1989; Arazi et al., 2004; Fifield et al., 2007; Miltenberger et al., 2017), the latter requires large accelerators ($\gtrsim$6 MV) and one may need to compromise the efficiency of ion transport and detection, which would reduce the advantage of the higher ionisation yield. An alternative emerging technique involves the use of lasers to suppress the $^{26}$MgO$^{-}$ ions after the ion source (Martschini et al., 2019; Lachner et al., 2019, 2021).

We use Al$^{-}$ for the measurement of $^{26}$Al and therefore the challenge almost entirely becomes optimising the ion source performance. Beyond the low ionisation efficiency of Al$^{-}$, there is the well known characteristic of ion source output decreasing during the course of a run. An example of this is provided in Fig. 2, which shows traces of $^{27}$Al$^{-}$ currents as a function of time from cathodes analysed during two separate runs. Traces in grey to black lines show a drastic reduction in output when we returned to them after a break off ∼60 hours that had been spent on measuring other samples that were part of the same run. When returned to these samples the output dropped to 200 – 300 nA despite the fact that the samples gave in the order of 800





– 1000 nA earlier. In contrast, the second set of samples (shown in shades of blue) analysed during a separate run shows no sign of the output deteriorating during the course of the run. The only notable difference between the two runs is the slightly different ion source parameters where the run with greater longevity (blue profiles) had lower sputtering voltage (4.5 kV vs. 6.5 kV) and a slightly lower Cs temperature.

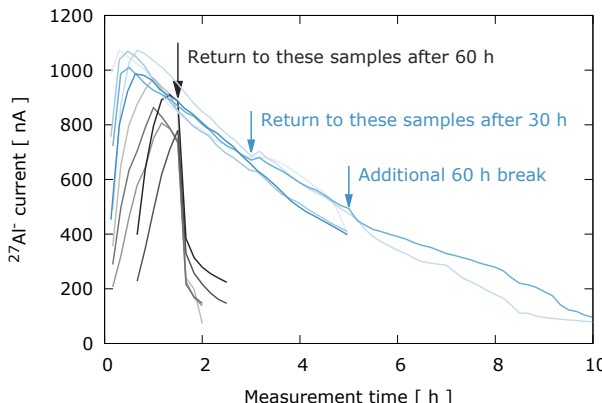

**Figure 2.** $^{27}$Al$^{-}$ output as a function of time for a representative set of samples from two different runs highlighting the idiosyncratic behaviour of Cs-sputter sources.

The difference in the source stability, as shown in Fig. 2, highlights the reason why it is often difficult to achieve the theoretical efficiency from real samples. Another practical issue that hinders the measurement of $^{26}$Al is that samples typically vary

in outputs when compared to standard reference materials. To quantify the latter, Fig. 3 shows histograms of sample outputs relative to the primary standard KN-4-2 (Nishiizumi, 2004) from three different sample preparation laboratories: ANSTO, University of Wollongong (UOW), and CosmIC Laboratories at Imperial College London (IMPERIAL).

Aluminium purification procedures at ANSTO and UOW follow Child et al. (2000) whereas those at CosmIC Laboratories mostly follow Corbett et al. (2016), respectively. On average samples prepared at ANSTO and UOW give outputs that are

60% and 50% of the primary standard, respectively. In contrast, samples prepared at CosmIC Laboratories at Imperial College London give on average 90% of the current of the standards but show the largest spread. $Al_2O_3$ from each laboratory is mixed with Ag at 1:2 ratio by weight and loaded into Cu cathodes. Whilst there are differences in how long the samples were measured we make the first order approximation that a set of samples from each laboratory would be subject to a similar spread. This allows us to quantify that the small differences in trace elemental composition between samples or differences in the process

of loading the samples, may yield a factor of two differences in outputs.

As in the case of $^{10}$Be, the second largest contributor to the $^{26}$Al measurement inefficiency is the charge state yield. The measured transmission for $^{27}$Al from 300 kV to 1 MV and from 4 MV to 6 MV accelerator voltages is shown in Fig. 4. Using Ar stripper gas, maximum transmissions to 1+, 2+ and 3+ charge states are 40%, 48%, and 40% at 0.5 MV, 1 MV, and 4 MV



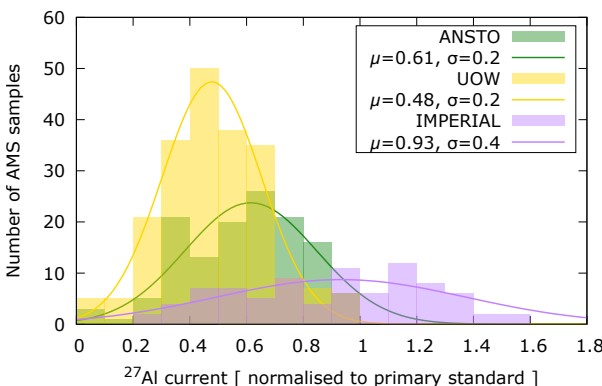

**Figure 3.** $^{27}Al^-$ output relative to the primary standard KN-4-2 (Nishiizumi, 2004) for collection of samples from three different sample preparation laboratories. Gaussian curves are fitted to the histograms with centroids and widths given.

accelerator voltages, respectively. Using He stripping, the maximum measured transmissions to 1+, 2+, and 3+ are 21%, 57%, and 39% at 0.6 MV, 0.5 MV, and 4 MV, respectively.

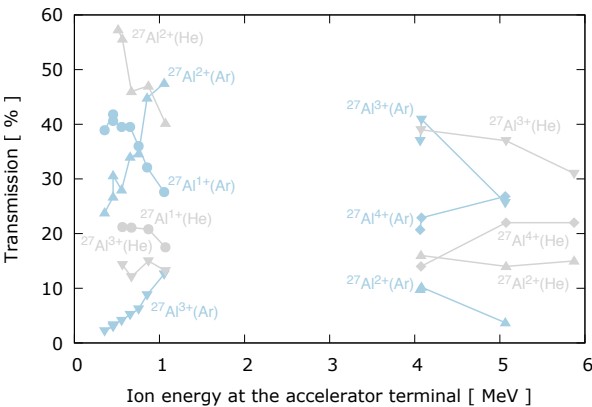

**Figure 4.** Compilation of $^{27}Al$ transmissions to various charge states as a function of ion energy for helium (grey) and argon (blue) stripper gases. The transmissions below 1 MeV ion energies were measured on VEGA and between 4 and 6 MeV on SIRIUS (Wilcken et al., 2015, 2017).

Our standard measurement setup uses 3+ charge state at 4 MV accelerator voltage with Ar stripper gas (Wilcken et al., 2019). The measured raw ratio of a standard material is typically around 90% of the reference value, which is due to a combination of transmission differences between $^{26}Al$ and $^{27}Al$, and losses in the ion detection. The above method does not utilise the maximum transmission that is achievable with 2+, but the benefit of clean spectra that is free of interferences has, to date,





out-weighted the losses in transmission. Equal ion optical transmission of about 40% can be achieved with 1+ charge state, but this reduces to about ∼30% by the time all the molecular interferences are sufficiently suppressed (Wilcken et al., 2015), and therefore offers no gain over the 3+ method.

The maximum transmission to 2+ is about 60% at 0.5 MV when using He stripper gas. Given the extremely low ionisation efficiency, the possibility to improve the measurement precision with 1.5 times higher count rates, when compared to our 170 standard 3+ method, is significant. However, the challenges with $^{26}Al^{2+}$ measurement are the interfering $^{13}C^{+}$ ions and how to separate these with low ion energies. Whilst the latter has been demonstrated elsewhere (e.g., Müller et al. 2015), we have not assessed the robustness of the method with our accelerator setup and samples from varying lithological settings.

### 2.3 Benchmarks

A simple approach to estimate the overall measurement efficiency is to look at the achieved precision from a suite of $^{10}Be$ 175 and $^{26}Al$ samples as shown in Fig. 5. These are minimum efficiency values because the samples are not fully consumed, and consumed to variable degrees. At isotope ratios below $10^{-13}$ the achievable precision becomes dominated by the counting statistics and from there on precision decreases for lower isotope ratios. At higher nuclide concentrations we typically limit the statistical precision to about 2% in measurement sequencing to allocate more time for low ratio samples and to ensure the measurement proceeds expeditiously. Reproducibility of standard reference materials is about 1% for both $^{10}Be$ and $^{26}Al$ as has 180 been demonstrated at different AMS laboratories: ANSTO (Wilcken et al., 2017), SUERC (Xu et al., 2015), and LLNL-CAMS (Rood et al., 2013).

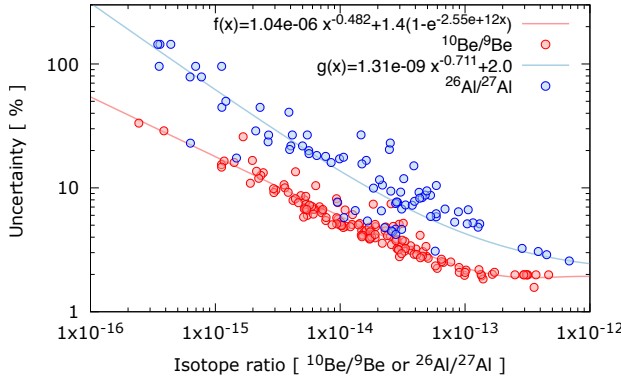

**Figure 5.** Measurement precision for a collection of $^{10}Be$ and $^{26}Al$ samples, as a function of $^{10}Be/^{9}Be$ and $^{26}Al/^{27}Al$ ratios. The fitted curves are given to allow easy estimation of achievable precision for future work and for inter-laboratory comparisons.

Fig. 5 demonstrates that it is possible to measure $^{26}Al$ and $^{10}Be$ to similar precision even for samples with isotope ratios at ∼ $10^{-15}$. This is possible for samples with $^{26}Al/^{10}Be$ ratios close to the surface production ratio of 6.75 (Balco et al., 2008), where the roughly factor of 10 lower ionisation efficiency for $Al^{-}$ compared to $BeO^{-}$ is compensated by the higher





concentration of $^{26}$Al atoms in the sample. However, currently the long measurement times required to achieve similar precision as for $^{10}$Be (see Fig. 2) reserve the capability only for a few selected projects, and for most samples the $^{26}$Al measurement precision is lower than for $^{10}$Be, as shown in Fig. 5.

A distinct feature of the $^{26}$Al samples shown in Fig. 5 is that the measurement precision is more variable than for $^{10}$Be and it is possible to see nearly a factor of 10 differences in measurement precision for samples with similar $^{26}$Al/$^{27}$Al ratios. This

is a direct consequence of either the variable output from samples as shown earlier in Fig. 3, or of the idiosyncratic behaviour of the ion source during the run as shown in Fig. 2. In contrast, $^{10}$Be behaves much more consistently which is reflected in the tight grouping of the data.

To allow a simple means to estimate achievable precision for future projects, functions have been fitted to the measured $^{10}$Be/$^{9}$Be and $^{26}$Al/$^{27}$Al data and are given in Fig. 5. The fitted functions were forced to result in measurement precision of

about 2% when ratios approach $1 \times 10^{-12}$. This is our routine measurement precision and whilst higher precision is possible for high-ratio samples, it is not pertinent here. The functions given in Fig. 5 are applicable to standard size samples with BeO and Al$_2$O$_3$ masses around 0.6 mg and 4 mg, respectively.

## 3  Background correction

With small signals, the importance of background correction is clear, and ultimately reduces to the question of when a signal

is above background (e.g., Currie, 1968). In AMS, one must employ Poisson statistics which is typically done as Gaussian approximation; however at very low number of counts, such approximation does not apply anymore (Currie, 1972; Schmidt et al., 1984; Feldman and Cousins, 1998). How differing measurement times between sample and blank will impact the lowest minimum detectable quantity has been discussed in Mathews and Gerts (2008), Potter and Strzelczyk (2008, 2011), and Alvarez (2013). However, before any such statistical analysis is undertaken, we argue that it is possible to isolate the most likely

source of background by fully characterising the AMS instrument in question with a suite of quality assurance samples. This information, in turn, will be our primary guide for selecting the most appropriate background correction method.

The background for $^{10}$Be or $^{26}$Al measurements can be categorised to arise from three distinct sources: (i) it may originate from the carrier used, (ii) it may be due to contamination during sample processing, and (iii) it may be due to instrument response, including sample cross talk in the ion source or insufficient suppression of interferences. Depending on the dominant

source of the background, the resulting blank correction scales differently. For example, if the dominant source is the carrier, the background correction should scale with the amount of carrier used; contamination during sample processing scales with the amount of reagents used or time in the laboratory; and if AMS measurement dominates the blank then the background correction should be based on the length of measurement and/or, in case of source memory, when the sample was measured, or the concentration of possible interfering isotopes. To account for all these different sources of background, one needs to

quantify their contribution and acknowledge that the most appropriate correction may differ between samples and nuclides. The latter is particularly the case with $^{26}$Al when one may or may not use carrier, or when carrier is used, different amounts may be added to each sample.





### 3.1 Background corrections for $^{10}$Be

The measured $^{10}$Be/$^9$Be ratio for an unknown sample and process blank can be written as:

$$R_{10,s} = \frac{N_{10,q} + N_{10,c}^s + N_{10,sp}^s + N_{10,AMS}^s}{N_{9,c}^s} \tag{1}$$

$$R_{10,b} = \frac{N_{10,c}^b + N_{10,sp}^b + N_{10,AMS}^b}{N_{9,c}^b} \tag{2}$$

where sub and superscripts $s$, $q$, $c$, $sp$, and $b$ refer to sample, quartz, carrier, sample process, and blank, respectively, and $R_{10,s}$ and $R_{10,b}$ are the measured $^{10}$Be/$^9$Be ratios in a sample and process blank, respectively. The number of $^{10}$Be atoms from the dissolved quartz is $N_{10,q}$. To account for potentially different amounts of carrier between samples and blanks these are labelled as $N_{10,c}^s$ and $N_{10,c}^b$, respectively. Similarly, the $^{10}$Be atoms that might enter the sample or blank during the sample processing are labelled as $N_{10,sp}^s$ and $N_{10,sp}^b$, respectively. The impact of AMS measurement on the recorded $^{10}$Be events is accounted with $N_{10,AMS}$ and again can be different between sample and blank. $N_{10,AMS}$ encapsulates sample cross-talk in the ion source as well as potential interference caused by high $^{10}$B rates. Equations (1) and (2) are based on the assumption that the amount of $^9$Be coming from the quartz, sample processing or AMS make negligible contributions as compared to the carrier itself. $^{10}$Be/$^9$Be ratios in the sample ($R_{10,s}$) and blank ($R_{10,b}$) are measured and the amounts of carrier added ($N_{9,c}^s$ and $N_{9,c}^b$) are known.

To be able to solve the number of $^{10}$Be atoms from the dissolved quartz from equations (1) and (2), one needs to either make assumptions about, or have external information on, the origin of the background $^{10}$Be events. The first common assumption is that the number of $^{10}$Be atoms that might enter a sample and a blank during sample processing is approximately equal, i.e.: $N_{10,sp}^s \approx N_{10,sp}^b$. This is a reasonable assumption in most circumstances as the sample preparation steps for unknown samples and blanks are closely mimicked.

The apparent $^{10}$Be counts observed due to the AMS instrument response ($N_{10,AMS}$) can be treated as equal between samples and blanks ($N_{10,AMS}^s \approx N_{10,AMS}^b$) under certain circumstances: they are measured close in time for similar duration when small source memory contribution can be approximated equal or there is no significant source memory, and B-rates are below a certain threshold above which the background $^{10}$Be/$^9$Be ratio starts to increase due to increasing interference rate. In Fig. 6, we plot a set of background test samples with artificially elevated boron concentrations (grey circles) along a representative collection of unknown samples (red circles) in order to illustrate the impact of increasing B-rates on measurement background. As shown in Fig. 6, samples located to the left of the inflection point in $^{10}$B test samples, where the measured $^{10}$Be/$^9$Be ratio starts to increase as a function of increasing interference rate do not need a separate B-correction. In this case, for example, all the unknown samples do not need a correction for B.

A further contribution of $^{10}$Be atoms in the blank is from the $^9$Be carrier itself. This may be quantified by preparing an AMS sample directly from the carrier by conversion of the carrier solution to BeO with minimal handling, and measuring it early in the run to avoid source memory build up. The measurable $^{10}$Be/$^9$Be ratio in the carrier can be written as $R_{10,c} = N_{10,c}/N_{9,c}$.

Now, we may solve the number of $^{10}$Be atoms in the quartz sample by rewriting equations (1) and (2):



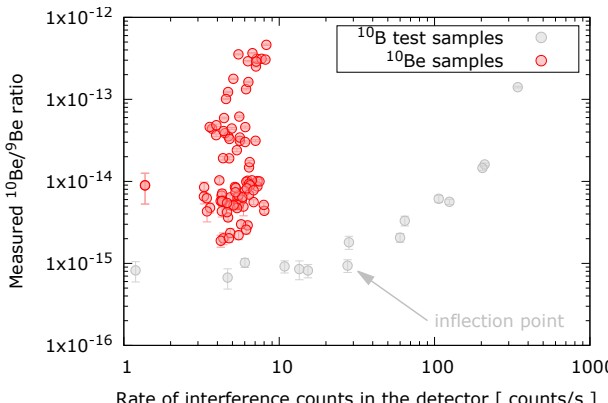

**Figure 6.** Compilation of $^{10}$B test samples (grey) and a collection of regular unknown samples (red) plotted against interference count rate in the detector. The inflection point where $^{10}$B induced interference starts to elevate the measured $^{10}$Be/$^9$Be ratio appears around 30 counts/s.

$$
\begin{aligned}
N_{10,q} &= [R_{10,s} - R_{10,c}] \times N_{9,c}^s \\
&- [R_{10,b} - R_{10,c}] \times N_{9,c}^b
\end{aligned}
\tag{3}
$$

The above is the general solution for background correction and requires one to measure both a process blank and a direct carrier blank. An obvious way to simplify the blank correction is to keep the carrier mass equal between a sample and a blank when the $R_{10,c}$ terms cancel out in equation (3) and we obtain:

$$
N_{10,q} = [R_{10,s} - R_{10,b}] \times N_{9,c}^s
\tag{4}
$$

Equation (4) is also reached if the dominant source of $^{10}$Be in the blank samples is the carrier itself, when $R_{10,b}$ approaches $R_{10,c}$ and the second term in equation (3) becomes very small. Note also that equation (4) is identical to the common background correction method where the number of atoms in the blank are scaled with the masses of the carrier solutions between samples and blank:

$$
\begin{aligned}
N_{10,q} &= R_{10,s} \times N_{9,c}^s - \frac{m_{9,s}}{m_{9,b}} \times R_{10,b} \times N_{9,c}^b \\
&= [R_{10,s} - R_{10,b}] \times N_{9,c}^s
\end{aligned}
\tag{5}
$$

where $m_{9,s}$ and $m_{9,b}$ are the masses of carrier solutions added to sample and blank, respectively.

In contrast, if the sample preparation process dominates the $^{10}$Be background, $R_{10,s}$ and $R_{10,b}$ become much larger than $R_{10,c}$ and equation (3) may be rewritten as:



$$N_{10,q} \quad \approx \quad R_{10,s} \times N_{9,c}^s - R_{10,b} \times N_{9,c}^b \tag{6}$$

Here it is noteworthy that when the sample preparation process itself is the dominant source of $^{10}$Be background atoms, the correction does not scale with the mass of the carrier in the sample as shown in equation (6). However, the simple practice of keeping the carrier masses equal between samples and blanks makes the blank correction very simple and one does not need to choose between equations (4) and (6) as they become identical. There may be situations when keeping carrier masses

between samples and blanks equal is not desired, however, as the carrier mass is proportional with the longevity of the sample during measurement and carrier masses may be chosen such that predicted $^{10}$Be/$^9$Be ratios are within a range that does not compromise the measurement.

### 3.2 Background corrections for $^{26}$Al

Similar to $^{10}$Be, the measured $^{26}$Al/$^{27}$Al ratio for an unknown sample $R_{26,s}$ and process blank $R_{26,b}$, respectively, can be

written as:

$$R_{26,s} \quad = \quad \frac{N_{26,q} + N_{26,c}^s + N_{26,sp}^s + N_{26,AMS}^s}{N_{27,q} + N_{27,c}^s} \tag{7}$$

$$R_{26,b} \quad = \quad \frac{N_{26,c}^b + N_{26,sp}^b + N_{26,AMS}^b}{N_{27,c}^b} \tag{8}$$

where $N_{26,q}$ is the number of $^{26}$Al atoms from the dissolved quartz, $N_{26,c}^s$ and $N_{26,c}^b$ are the numbers of $^{26}$Al atoms from the carrier, if any was used, in the sample and blank, respectively. Additional $^{26}$Al atoms that might enter the sample during sample

processing are $N_{26,sp}$, and the impact of AMS measurement to the recorded $^{26}$Al events is $N_{26,AMS}$. The latter encapsulates cross-talk in the ion source as well as any events that might have erroneously recorded as $^{26}$Al. As in the case of $^{10}$Be, contamination during processing or measurement ($N_{26,sp}$ and $N_{26,AMS}$) can be assumed to be equal between sample and blank as long as they are processed and measured in a similar way and there are no clear ion source memory effects or interferences that might be different between the two. The latter is quite a safe assumption when using Al$^-$ beams and the 3+

charge state for the measurement because the $^{26}$Al events are so clearly separated from any possible interferences. Equations (7) and (8) assume that possible $^{27}$Al contamination is negligible.

If the carrier $^{26}$Al/$^{27}$Al ratio ($R_{26,c} = N_{26,c}/N_{27,c}$) is known, we can solve the number of $^{26}$Al atoms from the dissolved quartz from equations (7) and (8):

$$N_{26,q} \quad = \quad R_{26,s} \times \left[ N_{27,q} + N_{27,c}^s \right]$$
$$\quad \quad - \quad R_{26,c} \times \left[ N_{27,c}^s - N_{27,c}^b \right]$$
$$- \quad R_{26,b} \times N_{27,c}^b \tag{9}$$





This general solution simplifies to a simple blank subtraction if the amounts of carrier added to the sample and blank are equal, in which case the second term drops out of equation (9). However, unlike in the case of $^{10}$Be, the presence of $^{27}$Al from the dissolved quartz means that the amounts of carrier added to the sample and blank are often different, and more commonly no
carrier is added at all to samples to bolster the $^{26}$Al/$^{27}$Al ratio. The above means that measurements of $R_{26,s}$, $R_{26,b}$, and $R_{26,c}$ are required in order to weigh the different sources of backgrounds appropriately.

Unfortunately, $^{26}$Al count rates are so low that it is often difficult to obtain a sufficient number of counts to be able to tell different blank samples apart. However, a collection of multiple blank measurements over a longer measurement time period offers higher statistical precision and allows differentiation of various blank samples, as shown in Table 1.

| Sample | N | $^{26}$Al [cnts] | $^{26}$Al-rate [cnts/s] | $\sigma(^{26}$Al-rate) [cnts/s] |
|---|---|---|---|---|
| Ag | 3 | 2 | $1.29 \times 10^{-4}$ | $9.13 \times 10^{-5}$ |
| Al-carrier | 5 | 22 | $9.14 \times 10^{-4}$ | $1.95 \times 10^{-4}$ |
| ANSTO | 29 | 151 | $2.31 \times 10^{-3}$ | $1.88 \times 10^{-4}$ |
| UOW | 25 | 91 | $1.10 \times 10^{-3}$ | $1.15 \times 10^{-4}$ |
| IMPERIAL | 22 | 118 | $8.85 \times 10^{-4}$ | $8.15 \times 10^{-5}$ |

**Table 1.** Long-term $^{26}$Al count rates from collection of blank samples: pure Ag powder, Al-carrier (1000 ppm ICP solution), and full process blanks from laboratories at ANSTO, University of Wollongong (UOW), and Imperial College London (IMPERIAL). Number of samples measured is N.

For $^{26}$Al measurements, we mix $Al_2O_3$ with Ag at a ratio of 1:2 by weight and load into Cu sample holders. To measure the ion source memory or the potential contribution from the Ag powder and/or Cu sample holder, we have measured the $^{26}$Al count rate from pure Ag powder with no $Al_2O_3$ present. As shown in Table 1, these 3 samples have had over 4 hours of cumulative sputtering and only 2 counts have been recorded yielding a count rate of $(1.3\pm0.9)\times10^{-4}$ counts/s. This is roughly an order of magnitude lower than what is measured from our Al-carrier (1000 ppm ICP solution), $(9.1\pm2.0)\times10^{-4}$ counts/s,
indicating that the source memory is not dominating the blank measurements. A similar conclusion is reached when comparing count rates for individual blanks that were measured at the beginning and at the end of a run with no clear difference. However, the low statistical precision presents a challenge [1].

When the full process blanks from the University of Wollongong and CosmIC Laboratories at Imperial College London are compared with the ICP carrier, the measured $^{26}$Al rates are within uncertainties, which indicates that the dominant source of

---

[1]On the other hand, one may argue, because there is no $Al_2O_3$ present in the pure Ag sample, the total output from the ion source is much less than from regular $Al_2O_3$ samples and may impact the probability of $^{26}$Al ions reaching the detector. Whilst this effect might be true, we believe the impact is within the uncertainties at these extremely low count rates.





$^{26}$Al in the blanks originates from the carrier itself in those laboratories. Conversely, full process blanks from ANSTO show roughly a factor of two higher count rate than the ICP carrier, indicating an additional source of $^{26}$Al in those samples.

A challenge with the general background correction equation (9) is that the $^{26}$Al/$^{27}$Al ratio in the carrier, $R_{26,c}$, might not be known with sufficient precision to improve the accuracy of the blank correction. Two end-member assumptions can be made to simplify equation (9). The first assumption is that the blank is dominated by the inherent $^{26}$Al in the carrier, which is the

likely scenario for the University of Wollongong and Imperial College London laboratories (see Table 1). In this case, $R_{26,b} \approx R_{26,c}$ and equation (9) simplifies to:

$$N_{26,q} \quad \approx \quad R_{26,s} \times \left[ N_{27,q} + N^s_{27,c} \right] - R_{26,b} \times N^s_{27,c} \tag{10}$$

The alternative assumption is that the dominant source of contamination is from sample processing. In this case, $R_{26,c}$ becomes small compared to $R_{26,b}$ and equation (9) can be written as:

$$N_{26,q} \quad \approx \quad R_{26,s} \times \left[ N_{27,q} + N^s_{27,c} \right] - R_{26,b} \times N^b_{27,c} \tag{11}$$

Equations (10) and (11) are similar to equations (4) and (6) for $^{10}$Be, with the same conclusion that if the carrier itself is the dominant source of $^{26}$Al or $^{10}$Be atoms in the blank sample, then the background correction is blank $^{26}$Al/$^{27}$Al ratio multiplied by the amount of carrier added to the sample. If no carrier is added to the sample, no blank correction is required. In contrast, if the dominant source of $^{26}$Al (or $^{10}$Be) atoms in the blank is sample processing, the background correction is calculated with

the amount of carrier added to the blank.

To demonstrate the impact of different background corrections – i.e., applying equations (9), (10), and (11) – we have corrected a set of samples with $^{26}$Al/$^{27}$Al ratios from $1 \times 10^{-15}$ to $1 \times 10^{-13}$ with the three different methods and results are shown in Fig. 7. The shaded area in Fig. 7 highlights the difference between (i) assuming that the source of background is the carrier itself and (ii) assuming that the main source of background is contamination during sample processing, i.e., equations

(10) and (11), respectively. General background correction using equation (9) is shown with blue circles. Fig. 7 demonstrates that for samples with $^{26}$Al/$^{27}$Al ratios above $1\times10^{-14}$ the choice of background correction is typically less significant, and all three methods lead to similar correction. In contrast, for samples with $^{26}$Al/$^{27}$Al ratios below $1\times10^{-14}$, not only can the magnitude of the background correction be similar to or larger than the measurement uncertainty (see Fig. 5) but also it can differ by up to 30% depending on which correction scheme is used. Therefore, knowing which correction method is the most

applicable to each case is highly pertinent for low ratio $^{26}$Al measurements.

The above sensitivity analysis differs philosophically from common approaches where varying blank values, e.g., the long-term mean or median, are subtracted to account for the effect of spurious high blank values (e.g., Corbett et al., 2017). Our approach addresses the uncertainty in the origin of the background events by using different correction methods. Using both approaches is recommended.





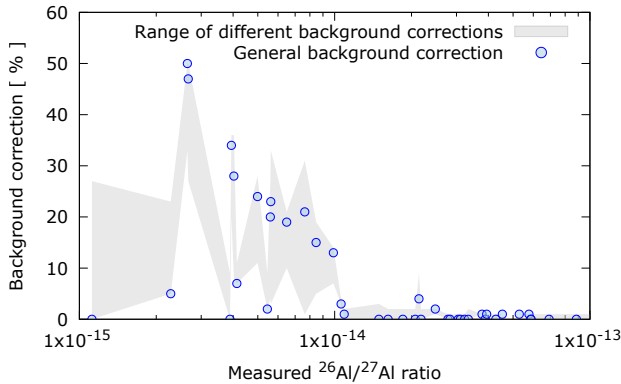

**Figure 7.** Impact of three different background corrections for a collection of $^{26}$Al samples as a function of $^{26}$Al/$^{27}$Al ratio. See text for more details.

## 4  Calculating isotope ratios

During an AMS measurement, a sample is typically measured between three to ten times for a few minutes each, where the number of repetitions and length of each measurement may vary depending on the AMS laboratory. These multiple measurements are then converted to a final result with an uncertainty that is either based on the number of rare isotope counts or that is spread between individual measurements (Elmore et al., 1984; Wacker et al., 2010). However, when the measured signal is small and the number of rare isotope counts is also low, the final result becomes sensitive to how the uncertainty is defined and how the individual measurements are combined. For discussions as how to calculate an uncertainty with low number of counts the reader is referred to earlier works (e.g., Currie, 1972; Schmidt et al., 1984; Feldman and Cousins, 1998).

To demonstrate the impact of different algorithms for calculating the final ratios and to validate our method, we show a sensitivity analysis based on a couple of low ratio $^{26}$Al samples. We selected five $^{26}$Al samples that were each measured $\sim$100 min during which 4 to 83 counts were recorded with an average $^{27}$Al$^{-}$ current of $(200-300)$ nA. To combine individual measurements to a final ratio, one may either sum the total counts and charge before calculating the final ratio or one may calculate a weighted mean from the individual measurement ratios. Whilst the former is independent of measurement sequence, the latter is easier to normalise for drifts in the spectrometer and is our preferred approach. To calculate weighted mean

$$R = \frac{\Sigma w_i r_i}{\Sigma w_i},\tag{12}$$

where $w_i$ is the weighing factor and $r_i$ is single isotope ratio measurement and R is the final ratio, for example Bevington and Robinson (2003). Following weighing factors are considered:

(i) *Equal weights*, where $w_i = 1$ and the final ratio is a simple arithmetic mean of the individual measurements;

(ii) *Integrated current*, where the weighing factor is the average $^{27}$Al current multiplied by measurement time;





(iii) *Uncertainty*, where $w_i = 1/\sigma_{Ai}^2$ and $\sigma_{Ai} = r_i X_i$, with $r_i$ being the measured ratio and relative uncertainty $X_i$ is assigned based on the number of $^{26}$Al counts $N_i$ during that run as shown in Table 2:

| $N_i$ | $X_i$ |
|---|---|
| 0 | 1.84 |
| 1 | 2.30 |
| 2 | 1.32 |
| $2 < N_i < 10$ | $(1+\sqrt{N_i})/N_i$ |
| $N_i \geq 10$ | $1/\sqrt{N_i}$ |

**Table 2.** Relative uncertainties used in the sensitivity analysis when the number of rare isotope counts is low, based on Schmidt et al. (1984).


(iv) *Normal uncertainty*, where $w_i = 1/\sigma_{Bi}^2$ and $\sigma_{Bi} = r_i/\sqrt{N_i}$, with $r_i$ and $N_i$ being the measured ratio and number of $^{26}$Al counts during that run, respectively.

Using the above weighing factors, we calculate the weighted mean of individual measurement ratios as a function of a single measurement duration for the selected five low ratio samples as shown in Fig. 8. Single measurement durations from 30 s to
$\sim$100 min were considered. To obtain nonzero weighing factors when these are based on uncertainties, one count was assumed in the case none were recorded.

Our sensitivity analysis (Fig. 8) demonstrates that the measured $^{26}$Al/$^{27}$Al ratio is dependent on the duration of the measurement if arithmetic mean or uncertainty based weighing factors are used to calculate the mean. Increasing the duration of a measurement will alleviate the differences until all the results converge to a single value that corresponds to a scenario where
all the data for a sample was collected during one long $\sim$100 min measurement. Differences in the measured $^{26}$Al/$^{27}$Al ratio and resulting isotope concentration can be 50% or more if only very few ($\lesssim$ 10) counts were recorded or about 30% if single measurement is shorter than 10 min. In contrast, if the integrated $^{27}$Al current is used to calculate the weighted mean, the final $^{26}$Al/$^{27}$Al ratio and subsequent nuclide concentration is independent of the measurement duration, as shown in Fig. 8.

To ensure that our results are independent of the measurement sequence, which is arbitrary in our case, our standard method
is to calculate the final isotope ratio by weighing the individual measurements with the integrated current as described in Wacker et al. (2010). However, as illustrated by our analysis, differing $^{26}$Al (or $^{10}$Be) concentrations may be obtained from the same data if alternate data reduction algorithms are used.

## 5 Conclusions

The challenge with measuring low $^{10}$Be or $^{26}$Al nuclide concentrations is to combine high AMS measurement efficiency with
low backgrounds. Achieving an increase in ionisation efficiency will not help if losses elsewhere in the instrument negate the

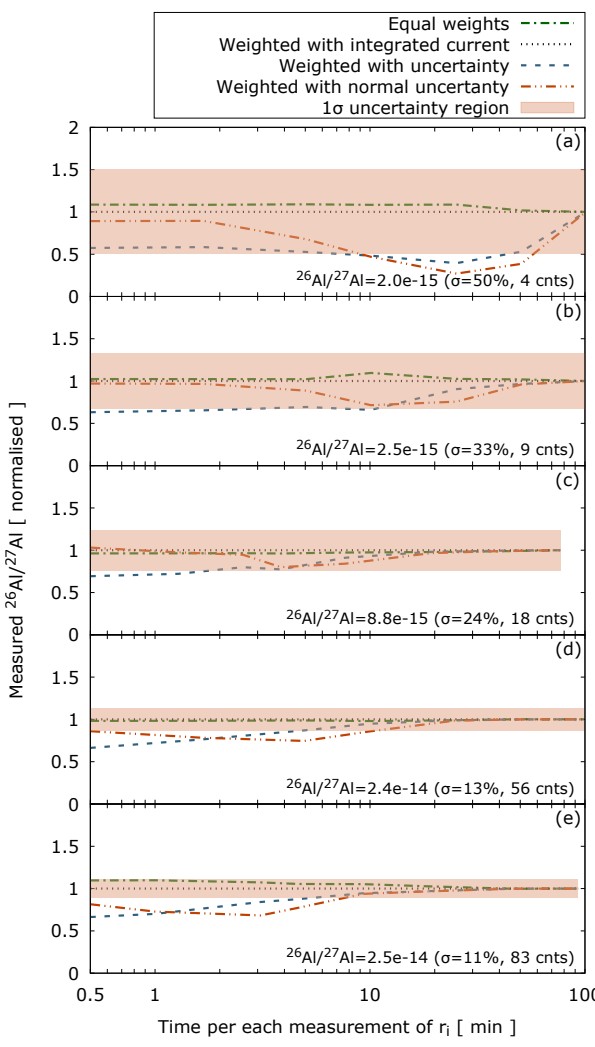

**Figure 8.** Impact of different methods for calculating the final $^{26}$Al/$^{27}$Al ratio for five low concentration samples as a function of the measurement duration. For each sample, we provide the total number of $^{27}$Al counts, uncertainty using Gaussian approximation, and $^{26}$Al/$^{27}$Al ratio.



gains. Similarly, achieving improved suppression of interferences in the ion detection is not sufficient if it comes at the cost of measurement efficiency. There is a strong connection between quality of AMS measurement and sample preparation, the latter being particularly important with low $^{10}$Be or $^{26}$Al nuclide concentrations. To facilitate a robust methodology for low nuclide concentrations, we recommend quality assurance practices that collate long-term data on carrier blank, ion source cross talk, and interference suppression. These practices add to the measurement time and cost but are essential to optimising analytical procedures that ultimately improve data quality and enable new applications.

By far the largest losses during an AMS measurement are in the ion source, where roughly 97% or 99.8% of the $^{10}$Be or $^{26}$Al atoms, respectively, are lost and do not contribute to the statistical precision. Our ion transmission through the accelerator for $^{10}$Be$^{3+}$ and $^{26}$Al$^{3+}$ are around 35% and 40%, respectively. In both cases, using the 2+ charge state would increase ion transmission to around 60% but whether this gain can be harnessed with equal or better backgrounds remains to be assessed. Using our measurement setup, we show that $^{26}$Al can be measured to similar precision as $^{10}$Be even for samples with $^{26}$Al/$^{27}$Al ratios in the range of $10^{-15}$, provided that measurement times are sufficiently long. For example, for an early Holocene sample with $^{10}$Be/$^9$Be and $^{26}$Al/$^{27}$Al ratios in the order of $5 \times 10^{-14}$ we would expect the uncertainties to be around 3 to 5%, whereas for a late Holocene sample with $^{10}$Be/$^9$Be and $^{26}$Al/$^{27}$Al ratios of $\sim 5 \times 10^{-15}$ we would expect the uncertainties to be in the range of 8 to 20%.

With low $^{10}$Be/$^9$Be or $^{26}$Al/$^{27}$Al ratios, the importance of background correction is obvious. However, this correction should not be done purely on the basis of process blanks, whether batch specific or long-term average, as the origin of the background events is an important factor in deciding the most appropriate correction method. For $^{26}$Al, for example, it is often not feasible to keep the mass of the carrier equal between samples and blanks, and the dominant source of the background events dictates the most appropriate blank correction. Our simple sensitivity analysis has shown that when $^{26}$Al/$^{27}$Al ratios are in the range of $(5 - 10) \times 10^{-15}$, typical for late Holocene surface exposure dating samples, or for burial dating samples, one can expect 30% differences in blank corrected concentrations and corresponding ages, depending on how blank corrections are done.

When the measured signal is small and the number of rare isotope counts is also low, the calculated final $^{10}$Be/$^9$Be or $^{26}$Al/$^{27}$Al ratio becomes sensitive to how the uncertainty is defined and how the individual measurements are combined. Our analysis demonstrates that the measured isotope ratio is dependent on the duration of the measurement, if arithmetic mean or uncertainty based weighing factors are used to calculate the mean. Differences in the resulting isotope concentration can be 50% or more if only very few ($\lesssim 10$) counts were recorded or about 30% if single measurement is shorter than 10 min. In contrast, if the integrated $^9$Be or $^{27}$Al current is used to calculate the weighted mean – our standard method – the final $^{10}$Be/$^9$Be or $^{26}$Al/$^{27}$Al ratio and subsequent nuclide concentration is independent of the measurement duration.

To summarise, our study presents a comprehensive method for analysis of cosmogenic $^{10}$Be and $^{26}$Al samples down to isotope concentrations of few thousand atoms per gram of sample, which opens the door to new and more varied applications of cosmogenic nuclide analysis.

*Data availability.* The data presented in this study is available by contacting the corresponding author.



*Author contributions.* KMW was responsible for the AMS analyses, ATC and RHF for the sample preparation at University of Wollongong,
KS and SK for the sample preparation at ANSTO, and AHR, DHR, and AJS for the sample preparation at Imperial College London. All
authors contributed to the writing of the manuscript.


*Competing interests.* The authors declare that they have no competing interests.

*Acknowledgements.* Financial support was provided by the University of Wollongong, Imperial College London, and the Centre for Accel-
erator Science at ANSTO through the National Collaborative Research Infrastructure Strategy (NCRIS). We acknowledge the Traditional
Custodians of the lands on which we have worked, and their continued spiritual and cultural connection to Country.






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
