# Peer review of "Technical note: Accelerator mass spectrometry of $^{10}$ Be and $^{26}$ Al at low nuclide concentrations"

_Geochronology, 2021_

## Author Response (AR1)

For brevity, we combine comments from both reviewers and for clarity we use different colours to indicate reviewer comments from author responses:

- Reviewer #1's comments are provided in red type
- Reviewer #2's comments are provided in blue type.
- Author responses are provided in black type.

Line 49: I do not believe that references are required for this line because AMS is an established technique by now. If references are to be added then I think some justification is needed for the inclusion of two references to the exclusion of all the other good reviews of the AMS process, or at minimum some acknowledgement that these are but two among many. While Finkel, Suter and Fifield are quite prominent in AMS and the cited texts are useful, there are similarly comprehensive reviews written by the original founders of the technique from the 1980s through 2010s that are available online (ex. at Mass Spectrometry Reviews). In particular, as Finkel and Suter (1993) is a book chapter, it is less accessible to many readers than accounts given in journal articles that can be downloaded online.

Given that a large part of the readership of the journal Geochronology are method users, we felt that including references here would be helpful for that part of the audience. To address reviewer's concern we added "e.g." before the references to make the point that these are some examples. We have also included Synal 2013 as reference.

Lines 51 – 54: the wording here is less precise than I think it should be. Magnets select for the momentum to charge ratio, electric analyzers for the kinetic energy to charge ratio, and there may be additional elements, such as a gas filled magnet or secondary stripper, to further help separate isobaric interferences. Statements such as, "... one mass with a given charge state ..." are not adequately descriptive. Secondly, neutrals can also be formed during electron stripping and molecular ions can survive the stripping process in low charge states if conditions are correct. Determining conditions to "kill" molecular interferences at low stripping energy for low charge states was key to the success of small AMS instruments.

We have modified this part and the new version now reads:

*"Negative ions are extracted from the sample via Cs sputtering (e.g., Middleton, 1983; Southon and Santos, 2007) and after electrostatic and magnetic analysis ions with selected mass to charge ratio are injected into an accelerator. In the accelerator terminal the ions undergo collisions with gas molecules (or a solid foil may also be used) that result in molecule break-up and the injected ions leaving the terminal with a range of charge states. A second electrostatic and magnetic analysis after the accelerator is used to select the mass to charge ratio for the final ion detection. "*

We consider items such as gas-filled-magnet as a part of ion detector and thus is not included in the simple overall statement discussed here.

"Section 2.1: Measurement of $^{10}$Be" needs major editing.

Figure 1: These data are confusing to me and I feel that they should not be accepted for publication as presented in the current form. They could be included as a guide for others if more clarification is provided, at least as outlined in 1 – 2 below.

1) Are these data being presented as equilibrium charge state yields?

1a) If not, then what these data represent needs to be more explicitly stated. If so, much more detail about the measurement process is needed, such as yield versus gas pressure and where the current was monitored. Although I suspect this not to be a major issue, what effort was taken to ensure that the beam composition measured at the accelerator injection was, in fact, nearly 100% that of the ion being investigated? For example, $^9Be^{16}O$ and $^{12}C^{13}C$ are molecular isobars. Was any effort made to look for such isobars at the high energy end? Or, was a target composed of the binding agent (Nb?) with no BeO inserted to see the injection current drop to 0?

1b) Transmissions are given as %, but no indication of the range of injection currents is given. I assume micro-Amperes, but some indication of a range would be useful.

2) The paragraph following the figure states that Be-, BeO- and $BeO_2$- were used to "cover a wide energy range".

2a) What do you mean by, "Ion energy at the accelerator terminal", then? Is this calculated as 9/original ion mass x terminal voltage? This is what line 100 appears to imply but I feel that methods to estimate ion energy should be more explicitly given in the text.

2b) How did you confirm that stripping yields are equivalent for molecular ions versus atomic ions? Which energy and charge state combination was used to normalize each of the data sets for molecular and atomic stripping yields? Why does the figure not indicate which data are for which ions? It is not clear to me that molecular ion stripping gives the same yield as atomic stripping, and no references are provided to back up this implied assumption. This would imply that any "coulomb explosion" associated with molecular break-up has no effect. It is known that there is a difference in the energy distribution of stripped molecular and atomic ions when stripping in foils.

2c) The figure caption states that, "The gap in the transmission data represents an energy

region where ion optical losses through the accelerator have greater impact on the measured charge states and so these data have been excluded". That sentence, to me, seems insufficient. How did you evaluate "optical losses" and how are these accounted for in the data that are presented?

> We modified this section most notably by replacing the charge state plot with a table of the most prominent charge states and transmissions that we can achieve. This is important point to the message of the paper and to allow the discussion on efficiency of the measurement and what options there are for us as well as other laboratories. Text was altered in few places in this section to refer to the table and streamline the discussion accordingly.

> By doing this we removed the need to go into discussion on "coulomb explosion" or other charge state measurement details, such as atomic vs molecular ions, that we felt would distract the reader from the message of the paper.

Lines 103-104: It might be useful to state where the charge state peak is for 3+. Also, this sentence is slightly confusing. Are the transmissions ~35% and ~18% at the 3+ peak or are these just the efficiencies at the max energy for 6 MV acceleration voltage?

We can't reach the peak with SIRIUS that is a 6 MV accelerator and there is limited number of measurements that would actually go clearly over the peak, particularly for He. Using ANTARES, an older accelerator at ANSTO, we still get similar 35% transmission at ~ 8 MV for Ar due to ion optical losses. We added "measured" to the sentence to clarify that the given transmission values were at 6 MV not theoretically at the peak.

Line 109: I am slightly confused by the idea that the raw ratio is 80-90% of the reference value. Isn't the raw ratio in units of counts/nC? Presumably the total charge and charge state was used to convert to atoms 9Be, but this should be made clear since raw ratio might be incorrectly interpreted. This phrasing is also used in Line 162.

"raw ratio" was changed to "unnormalised 10Be/9Be ratio" to clarify the point.

Section 2.2:

Lines 135 – 142: The phenomenon of current fall-off has been observed by others, but it is not fully understood and if these data are presented then clarification is needed. Is this a regular occurrence for you or was the 6.5kV run a "one-off"? Are you suggesting that all targets that are idle for 60 hours after initial sputtering at 6.5kV will under-perform as yours did? Klein and Mous (2017) NIMB 406, 210-213 suggest using > 11kV, and Southon and Roberts (2000) NIMB 172, 257-261 also suggest higher target voltages. The standard sputter voltage for the HVE SO110b is 7kV for Al2O3 + Ag targets. While this current drop-off effect has been observed by others, are you suggesting that it is solely due to the sputter voltage and could not have been due to other factors? The 6.5kV targets had a slow rise and the largest peak current output from that "grey" set was less than the least peak current output from the 4.5kV "blue" set. For example, are you certain that your Al2O3 : Ag ratios were correct and that the pins were all firmly pressed? Was the ion source cleaned prior to each of the 6.5kV and 4.5kV runs? The authors need to clearly state how certain they are that this current drop was indeed due only to the sputter voltage and how they arrived at that conclusion. Otherwise, they should re-word this to avoid further confounding an already confusing issue.

Lines 135-142: This is super interesting! Is the only difference between the blue and black current trends really just the cathode voltage? I suspect the authors also had to reposition the target with respect to the ionizer to optimize the Cs focus between these settings. Maybe that doesn't matter for the ANSTO setup though? I think it would be useful to know if the authors examined the cathodes after the analysis and noticed any differences in sputter style between 6.5 and 4.5 kV.

As noted by both reviewers this part was clearly ambiguous and we have clarified the point of the graph by additional reference to earlier work by Middleton that suggested higher voltages and clarifying our position in regards to that. The relevant part now reads:

*"Whilst earlier work by Middleton (1983) showed that higher sputtering voltages yield an increase in Al⁻ output, we observed that this might sometimes be overshadowed by deterioration in the source output as with 4.5 kV sputtering voltages (blue profiles) in Fig. 1 the samples performed much more consistently as a function of time than with 6.5 kV sputtering voltages (grey profiles). However, whilst we are confident that the difference in longevity of the source is not simply a matter of quality in target preparation/packing, as the feature existed across multiple samples from different sources, we are not suggesting that the increased longevity is solely due to lower sputtering voltage. We cannot control all the parameters, such as the amount of Cs in the source between experiments and therefore cannot isolate the cause for the observed improvement in longevity. "*

Despite inconclusive result we believe it is important to make this point part of the manuscript as a means to encourage others to experiment with new ideas.

Lines 148 – 155: Lines 151-2 were a little unclear: Were all the targets mixed with Ag and pressed into cathodes at ANSTO or at the labs where the Al2O3 was prepared? Were the Al2O3 powders analyzed for actual %[Al] content by any methods other than AMS? Did all the Al2O3 powders have a pristine color? Does each lab have a similar [Al] yield from the sample preparation procedures?

Lines 151-154: It would be useful to know if there are any specific differences in the prep methods between labs that might explain such significant differences in output. Perhaps looking at a subset of blanks would at least control for elemental impurities.

We agree that there are a lot of details that could explain this, but unfortunately, we don't have all the answers yet. Samples with lower yields in Al will not last as long but to first order will give out similar currents initially. We are currently in the process of collecting more information on this with hope to shed more light here and to achieve better consistency. The sample packing was done independently at each laboratory and have now been clarified in the text.

Figure 4: As was the case for Figure 1, I do not feel that these data should be accepted for publication as currently presented. Similar clarifications as those outlined for Figure 1 listed above should be given.

We followed the approach taken with Figure 1 that got replaced with a table with transmission for given charge states and replaced original Figure 4 with Table 2. Similarly, to the case of Be we believe this avoids the need to get into the details of charge state measurements that would distract from the central message of the paper. As with Be term "raw ratio" was replaced with *"measured unnormalised $^{26}Al/^{27}Al$ ratio".*

Lines 182-187: I get the point the authors are trying to make here with the higher production rate in quartz compensating for the 10x poorer ionization efficiency. However, there is the additional complication of the 26Al/27Al ratio being fundamentally limited by the native 27Al in the quartz, which somewhat dampens the point they are trying to make here. Also, the authors should note that this surface production ratio is specific to quartz.

To clarify, notion about the production rate in quartz was added. Yes, we agree Be and Al systematics are different and there are complications but in order to have some predictive capability we believe it is valuable to make generalisation that is applicable in many cases.

Equation 1, line 220: I disagree with this equation. First, the denominator does not include 9Be from the sample material itself. Is the sample material assumed to be quartz? Or is the "dissolved quartz" in line 224 assumed to be contamination from the quartz vessels used to process the original material from which Be is to be extracted? If quartz is the assumed sample material then the denominator in equation 1 should read exactly as the numerator reads, but with subscript "10" replaced by "9". An explicit statement should then be added that, very reasonably, the assumption is made that $N^S_{9,sp} + N^S_{9,AMS} \ll$

$N^S_{9,q} + N^S_{9,c}$ so that those terms are ignored. However, in present form, exclusion of the $N^S_{9,q}$ term from equation 1 affects the following equations 3 – 6. This section should be almost identical to that for 26-Al, as far as I can tell. For example, in the current form, the 10Be/9Be ratio approaches

infinite as the amount of carrier added to the sample approaches zero. This does not appear to be a typo as equation 3 follows from equations 1 and 2 in their current forms. This, then, renders the remainder of the section in error.

Equations 1 & 2 were based on assumption that intrinsic 9Be from dissolved quartz, or from sample processing and AMS measurement (e.g. target holder) is much smaller than from the carrier. This was stated in the original manuscript but the point is now further clarified by stating that Eqs 1 & 2 are approximations before writing them. We believe this approach is valid as it is most relevant to majority of readers and for those who want to do carrier free measurements the equations indeed follow 26Al as the reviewer stated, but since this is currently not the norm we believe the presented method is justified.

Lines 240-242: Please describe how the 10B test samples were artificially elevated. Diluted drops of boric acid added to a carrier solution before final hydroxide precipitation?

The following sentence was added to the text to clarify the matter:

"The $^{10}B$ test samples were prepared by adding diluted 1000 ppm B standard solution to crucibles with $^9Be$ carrier solution, the mixture being subsequently dried down and calcined."

Line 248: This makes me curious about what sort of source memory build up is typically observed. For example, what is "early in the run"? Also, was this effect considered for the later experiments looking at the Ag and carrier blanks? Some further detail here would be relevant.

This is more of a precaution than anything else. Typically, there is no notable difference between Ag (or Nb in case of Be) and carrier blanks measured at the beginning and at the end. Of course, count rates are low and in case of high ratio samples this might become relevant. Thus, we didn't want to make overarching statements regarding "no memory is observed".

Lines 269-272: One consideration that might be added to the discussion, perhaps in this section(?), is that distinguishing the contribution of "sample process" and "carrier" atoms could be done by analyzing process blanks with different carrier masses. Plotting measured atoms vs. mass of carrier added, then fitting a line, should give you both.

We agree, but the trouble is the low measurement precision and that sample contamination during processing won't be constant and thus needs to be established for each batch of samples. Therefore, measuring end-members regularly is a more cost-effective method. However, this could and should be done as one-off experiment.

Lines 253 – 254: The sentence needs to edited. As it is written, the sentence needs to be split at line 254 "when" and then the two sentences edited accordingly. However, it is a good idea to state the implicit assumption that if the carrier masses are roughly equal than the two "N9,c" rare isotope count terms are roughly equal for "s" and "b". You have stated above that Al- current yields from different labs are not the same so the reader should be alerted to the possible pit-fall of this assumption.

The first point was clarified. As for the second point samples and blanks should not be mixed between labs and thus differences in yields are not critically pertinent here, unless we have misunderstood the reviewer's intentions.

Lines 350 – 355: I have also been thinking about this issue for some time. If we "sum the total counts and charge before calculating the final ratio" and we assume instrumental drift is cyclical over a period of time comparable to the full measurement time of a target then we are actually better off just burning through each target in sequence rather than separating the analyses into shorter intervals.

There are many ways to do this but important point here is to be aware of the potential pit-fall. Hopefully we as community reach a point where data reduction algorithms are openly available and transparent at some point.

Technical Corrections

Line 87: This sentence needs editing for grammar. For example, "Practical methods to optimize sample consumption and negative ionization include: ... ii) what binding material is used...." should be reworded.

This was changed and now reads:

*"Practical methods to optimise sample consumption and negative ionisation include: (i) optimising the shape of the sample holder, (ii) changing the position that the sample is loaded in relative to the sample holder, (iii) using different binding materials and amounts relative to the amount of sample material, (iv) optimising the position of the sample within the ion source, and (v) finding the optimal operating conditions for each ion source."*

Lines 210 – 214: For clarity, this sentence should be split into smaller sentences rather than using semicolons to separate ideas.

Lines 214-217: Yes!!

Here we had contradictory views by reviewers and given reviewer 1 was happy with this sentence we left it as is.

Line 222: "where sub and superscripts …" >> "where sub- and superscripts…" or "where subscripts and superscripts…"

Amended.

Line 226: Poor grammar > "The impact of AMS measurement on the recorded 10Be events is accounted with N10,AMS and again can be different between sample and blank."

This has been changed and the sentence now reads:

*"Any $^{10}$Be events that are recorded due to the AMS measurement itself, are accounted with $N_{10,AMS}$ and again can be different between sample and blank."*

Lines 237 – 240: Grammar > "... close in time for similar duration when small source memory contribution can be approximated equal…". I also suggest splitting the sentence into two sentences.

This has been changed and the sentence now reads:

*"The apparent $^{10}$Be counts observed due to the AMS instrument response ($N_{10,AMS}$) can be treated as equal between samples and blanks ($N^s_{10,AMS} \approx N^b_{10,AMS}$) under the following circumstances: samples and blanks are measured close in time for a similar duration when small source memory contribution can be approximated equal, and B-rates are below a certain threshold above which the background $^{10}Be/^9Be$ ratio starts to increase due to increasing interference rate."*

Lines 310-311: Are there significant differences in current between process blanks and carrier blanks? Probably not twice as much. Also, the currents would likely be lower for the process blank (opposite what would be needed to explain the count rate difference with higher currents). However, this is important to the conclusion drawn here so some comment on current similarity would be useful.

Text has been clarified with following sentence added:

*"Ion source output between Al-carrier and full chemistry blanks is roughly equal with no systematic differences between the two. "*

Lines 346 - 347: The grammar needs editing.

This has been changed and the sentence now reads:

*"For more information on how to calculate an uncertainty when the number of counts are low, the reader is referred to earlier works (e.g., Currie, 1972; Schmidt et al., 1984; Feldman and Cousins, 1998). "*

Lines 372-373: This, along with Figure 8., is profound and cool! Weighting by total charge makes sense, but I would not have expected the other methods to be so poor.

Thanks, we agree.

Figure 1: The light grey used for the data and text annotation in the plot made it slightly difficult to read on screen and extremely difficult on a printout. I recommend using a higher contrast color. This also applies to Figure 6.

Figure 1 was removed and we changed the light-grey to darker grey in now Figure 4 (was Figure 6).

Figure 5: It would be interesting to also see where the theoretical best curves are—that is, what counting statistics one could get if a target was exhausted vs. ratio.

We agree in principle but for clarity and brevity we chose not to include this due to complications discussed earlier related to this figure, namely variable amount of native 27Al in quartz and potentially adding carrier. To calculate theoretical curves would be based on assumptions on the sample masses and whilst we could use similar masses as our typical samples, we feel the curves could potentially be misinterpreted as the limit of the technique rather than a snap-shot of the current performance under certain assumptions. The important message of the figure is the predictive capability for typical samples. Secondary

question as how far this is from current theoretical performance is left for the reader. All the relevant values are given to perform this calculation.

Table 1: For clarity, you might replace (26Al [cnts]) with (26Al [tot cnts]) since this is the sum of all counts measured over N targets.

Thank you, we have now modified the text accordingly.

---

## Author Response (AR2)

Dear editor,

Thank you for accepting our manuscript.

We have clarified the section around line 196 as requested with the suggested edit.

Regarding the second point as whether n or N should be used we decided to keep using N. Reason behind our decision was that whilst consistency across literature would be great unfortunately we won't be able to solve the matter here whichever nomenclature we use.

Best Regards,
Klaus